# Newton–Cotes Graph Neural Networks:
# On the Time Evolution of Dynamic Systems

**Lingbing Guo**[1,2,3]*, **Weiqing Wang**[4]*, **Zhuo Chen**[1,2,3], **Ningyu Zhang**[1,2],
**Zequn Sun**[5], **Yixuan Lai**[1,2,3], **Qiang Zhang**[1,3]† **and Huajun Chen**[1,2,3]†
[1]College of Computer Science and Technology, Zhejiang University
[2]Zhejiang University - Ant Group Joint Laboratory of Knowledge Graph
[3]ZJU-Hangzhou Global Scientific and Technological Innovation Center
[4]Department of Data Science & AI, Monash University
[5]State Key Laboratory for Novel Software Technology, Nanjing University

## Abstract

Reasoning system dynamics is one of the most important analytical approaches for many scientific studies. With the initial state of a system as input, the recent graph neural networks (GNNs)-based methods are capable of predicting the future state distant in time with high accuracy. Although these methods have diverse designs in modeling the coordinates and interacting forces of the system, we show that they actually share a common paradigm that learns the integration of the velocity over the interval between the initial and terminal coordinates. However, their integrand is constant w.r.t. time. Inspired by this observation, we propose a new approach to predict the integration based on several velocity estimations with Newton–Cotes formulas and prove its effectiveness theoretically. Extensive experiments on several benchmarks empirically demonstrate consistent and significant improvement compared with the state-of-the-art methods.

## 1 Introduction

Reasoning the time evolution of dynamic systems has been a long-term challenge for hundreds of years [1, 2]. Despite that the advances in computer manufacturing nowadays make it possible to simulate long trajectories of complicated systems constrained by multiple force laws, the computational cost still imposes a heavy burden on the scientific communities [3–8].

Recent graph neural networks (GNNs) [9–13]-based methods provide an alternative solution to predict the future states directly with only the initial state as input [14–20]. Take molecular dynamics (MD) [4, 21–24] as an example, the atoms in a molecule can be regarded as nodes with different labels, and thus can be encoded by a GNN. As the input and the target are atomic coordinates, the learning problem becomes a complicated regression task of predicting the coordinate at each dimension of each atom. Furthermore, some directional information (e.g., velocities or forces) are also important features and cannot be directly leveraged especially considering the rotation and translation of molecules in the system. Therefore, the existing works put great effort into the reservation of physical symmetries, e.g., transformation equivariance and geometric constraints [18, 19]. The experimental results on a variety of benchmarks also demonstrate the advantages of their methods.

Without loss of generality, the main target of the existing works is to predict the future coordinate $\mathbf{x}^T$ distant to the given initial coordinate $\mathbf{x}^0$, with $\mathbf{x}^0$ and the initial velocity $\mathbf{v}^0$ as input. Then, the

---

*Equal Contribution

†Correspondence to: {qiang.zhang.cs, huajunsir}@zju.edu.cn,

predicted coordinate $\hat{\mathbf{x}}^T$ can be written as:

$$\hat{\mathbf{x}}^T = \mathbf{x}^0 + \hat{\mathbf{v}}^0 T. \tag{1}$$

For a complicated system comprising multiple particles, predicting the velocity term $\hat{\mathbf{v}}^0$ rather than the coordinate $\hat{\mathbf{x}}^T$ improves the robustness and performance. Specifically, by subtracting the initial coordinate $\mathbf{x}^0$, the learning target can be regarded as the normalized future coordinate, i.e., $\hat{\mathbf{v}}^0 = (\mathbf{x}^T - \mathbf{x}^0)/(T - 0)$, which is why all state-of-the-art methods adopt this strategy [18, 19]. Nevertheless, the current strategy still has a significant drawback from the view of numerical integration.

As illustrated in Figure 1, supposed that the actual velocity of a particle is described by the curve $v(t)$. To predict the future state $\mathbf{x}^T$ at $t = T$, the existing methods adopt a constant estimation approach, i.e., $\hat{\mathbf{x}}^T = \mathbf{x}^0 + \hat{\mathbf{v}}^0 T = \mathbf{x}^0 + \int_0^T \hat{\mathbf{v}}^0 dt$. Evidently, the prediction error could be significant as it purely relies on the fitness of neural models. If we alternatively choose a simple two-step estimation (i.e., Trapezoidal rule [25]), the prediction error may be reduced to only the blue area. In other words, the model only needs to *compensate* for the blue area.

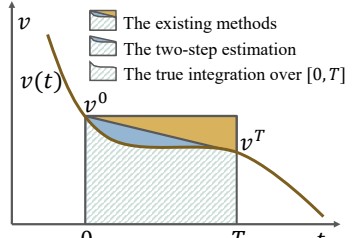

Figure 1: Illustration of different estimations. The rectangle, trapezoid, and striped areas denote the basic (used in the existing works), two-step, and true estimations, respectively. Blue denotes the error of two-step estimation that a model needs to compensate for, whereas blue + yellow denotes the error of the existing methods.

In this paper, we propose Newton–Cotes graph neural networks (abbr. NC) to estimate multiple velocities at different time points and compute the integration with Newton-Cotes formulas. Newton-Cotes formulas are a series of formulas for numerical integration in which the integrand (in our case, $v(t)$) are evaluated at equally spaced points. One most important characteristic of Newton-Cotes formulas is that the integration is computed by aggregating the values of $v(t)$ at these points with a group of weights $\{w^0, ..., w^k\}$, where $k$ refers to the order of Newton–Cotes formulas and $\{w^0, ..., w^k\}$ is irrelevant to the integrand $v(t)$ and can be pre-computed by Lagrange interpolating polynomial [26].

We show that the existing works can be naturally derived to the basic version NC ($k = 0$) and theoretically prove that the prediction error of this estimation as well as the learning difficulty will continually reduce as the increase of estimation step $k$.

To better train a NC (k), we may also need the intermediate velocities as additional supervised data. Although these data can be readily obtained (as the sampling frequency is much higher than our requirement), we argue that the performance suffers slightly even if we do not use them. The model is capable of learning to generate promising velocities to better match the true integration. By contrast, if we do provide these data to train a NC (k), denoted by NC$^+$ (k), we will obtain a stronger model that not only achieves high prediction accuracy in conventional tasks, but also produces highly reliable results of long-term consecutive predictions.

We conduct experiments on several datasets ranging from N-body systems to molecular dynamics and human motions [27–29], with state-of-the-art methods as baselines [17–19]. The results show that NC improves all the baseline methods with a significant margin on all types of datasets, even without any additional training data. Particularly, the improvement for the method RF [17] is greater than 30% on almost all datasets.

## 2   Related Works

In this section, we first introduce the related works using GNNs to learn system dynamics and then discuss more complicated methods that possess the equivariance properties.

**Graph Neural Networks for Reasoning Dynamics**   Interaction network [30] is perhaps the first GNN model that learns to capture system dynamics. It separates the states and correlations into two different parts and employs GNNs to learn their interactions. This progress is similar to some simulation tools where the coordinate and velocity/force are computed and updated in an alternative fashion. Many followers extend this idea, e.g., using hierarchical graph convolution [9, 10, 31, 32] or auto-encoder [33–36] to encode the objects, or leveraging ordinary differential equations for energy

conservation [37]. The above methods usually cannot process the interactions among particles in the original space (e.g., Euclidean space). They instead propose an auxiliary interaction graph to compute the interactions, which is out of our main focus.

**Equivariant Graph Neural Networks**   Recent methods considering physical symmetry and Euclidean equivariance have achieved state-of-the-art performance in modeling system dynamics [14–19, 38]. Specifically, [14–16] force the translation equivariance, but the rotation equivariance is overlooked. TFN [39] further leverages spherical filters to achieve rotation equivariance. SE(3) Transformer [29] proposes to use the Transformer [13] architecture to model 3D cloud points directly in Euclidean space. EGNN [18] simplifies the equivariant graph neural network and make it applicable in modeling system dynamics. GMN [19] proposes to consider the physical constraints (e.g., sticks and hinges sub-structures) widely existed in systems, which achieves the state-of-the-art performance while maintaining the same level of computational cost. SEGNN [38] extends EGNN with steerable vectors to model the covariant information among nodes and edges. Note that, not all EGNNs are designed to reason system dynamics, but the best-performing ones (e.g., EGNN [18] and GMN [19]) in this sub-area usually regard the velocity $\mathbf{v}$ as an important feature. These methods can be set as the backbone model in NC, and the models themselves belong to the basic NC (0).

**Neural Ordinary Differential Equations**   The neural ordinary differential equation (neural ODE) methods [40, 41] parameterize ODEs with neural layers and solve them by numerical solvers such as 4th order Runge-Kutta. These solvers are originally designed for numerical integration. Our work draws inspiration from numerical integration algorithms, where the integrand is known only at certain points. The goal is to efficiently approximate the integral to desired precision. Neural ODE methods repeatedly perform the numerical algorithms on small internals to obtain the numerical solution. Therefore, it may be feasible to iteratively perform our method to solve problems in neural ODEs.

## 3   Methodology

We start from preliminaries and then take the standard EGNN [18] as an example to illustrate how the current deep learning methods work. We show that the learning paradigm of the existing methods can be derived to the simplest form of numerical integration. Finally, we propose NC and theoretically prove its effectiveness in predicting future states.

### 3.1   Preliminaries

We suppose that a system comprises $N$ particles $p_1, p_2, ..., p_N$, with the velocities $\mathbf{v}_1, \mathbf{v}_2, ..., \mathbf{v}_N$ and the coordinates $\mathbf{x}_1, \mathbf{x}_2, ..., \mathbf{x}_N$ as states. The particles can also carry various scalar features (e.g., mass and charge for atoms) which will be encoded as embeddings. We follow the corresponding existing works [17–19] to process them and do not discuss the details in this paper.

**Reasoning Dynamics**   Reasoning system dynamics is a classical task with a very long history [1]. One basic tool for reasoning or simulating a system is numerical integration. Given the dynamics (e.g., Langevin dynamics, well-used in MD) and initial information, the interaction force and acceleration for each particle in the system can be estimated. However, the force is closely related to the real-time coordinate, which means that one must re-calculate the system states for every very small time step (i.e., $dt$) to obtain a more accurate prediction for a long time interval. For example, in MD simulation, the time step $dt$ is usually set to 50 or 100 fs (1 fs = $10^{-15}$ second), whereas the total simulation time can be 1 ms ($10^{-3}$ second). Furthermore, pursuing highly accurate results usually demands more complicated dynamics, imposing heavy burdens even for super-computers. Therefore, leveraging neural models to directly predict the future state of the system has recently gained great attention.

### 3.2   EGNN

Although the existing EGNN methods have great advantage in computational cost over the conventional simulation tools, the potential prediction error that a neural model needs to compensate for is also huge. Take the standard EGNN [18] as an example, the equivariant graph convolution can be written as follows:

$$\mathbf{m}_{ij}^0 = \phi_e(\mathbf{h}_i, \mathbf{h}_j, ||\mathbf{x}_i^0 - \mathbf{x}_j^0||^2, e_{ij}), \tag{2}$$

where the aggregation function $\phi_e$ takes three types of information as input: the feature embeddings $\mathbf{h}_i$ and $\mathbf{h}_j$ for the input particle $p_i$ and an arbitrary particle $p_j$, respectively; the Euclidean distance $||\mathbf{x}_i^0 - \mathbf{x}_j^0||^2$ at time $t = 0$; and the edge attribute $e_{ij}$. Then, the predicted velocity and future coordinate can be calculated by the following equation:

$$\hat{\mathbf{v}}_i^0 = \phi_v(\mathbf{h}_i)\mathbf{v}_i^0 + \frac{1}{N-1}\sum_{j \neq i}(\mathbf{x}_i^0 - \mathbf{x}_j^0)\mathbf{m}_{ij}^0, \tag{3}$$

$$\hat{\mathbf{x}}_i^T = \mathbf{x}_i^0 + \hat{\mathbf{v}}_i^0 T, \tag{4}$$

where $\phi_v : \mathbb{R}^D \to \mathbb{R}^1$ is a learnable function. From the above equations, we can find that the predicted velocity $\hat{\mathbf{v}}_i^0$ is also correlated with three types of information: 1) the initial velocity $\mathbf{v}_i^0$ as a basis; 2) the pair-wise Euclidean distance $||\mathbf{x}_i^0 - \mathbf{x}_j^0||^2$ (i.e., $\mathbf{m}_{ij}^0$) to determine the amount of force (analogous to Coulomb's law); and the pair-wise relative coordinate difference $(\mathbf{x}_i^0 - \mathbf{x}_j^0)$ to determine the direction. For multi-layer EGNN and other EGNN models, $\hat{\mathbf{v}}_i^0$ is still determined by these three types of information, which we refer interested readers to Appendix A for details.

**Proposition 3.1** (Linear mapping). *The existing methods learn a linear mapping from the initial velocity $\mathbf{v}^0$ to the average velocity $\mathbf{v}^{t^*}$ over the interval $[0, T]$.*

*Proof.* Please see Appendix B.1 □

As the model makes prediction only based on the state and velocity at $t = 0$, we assume that $\mathbf{v}^{t^*}$ fluctuates around $\mathbf{v}^0$ and follows a normal distribution denoted by $\mathcal{N}_{NC(0)} = (\mathbf{v}^0, \sigma_{NC(0)}^2)$. Then, the variance term $\sigma_{NC(0)}^2$ reflects how difficult to train a neural model on data sampled from this distribution. In other words, the larger the variance is, the worse the expected results are.

**Proposition 3.2** (Variance of $\mathcal{N}_{NC(0)}$). *Assume that the average velocity $\mathbf{v}^{t^*}$ over $[0, T]$ follows a normal distribution with $\mu = \mathbf{v}^0$, then the variance of the distribution is:*

$$\sigma_{NC(0)}^2 = \mathcal{O}(T^2) \tag{5}$$

*Proof.* According to the definition, $\sigma_{NC(0)}^2$ can be written as follows:

$$\sigma_{NC(0)}^2 = \frac{\sum_p (\mathbf{v}^{t^*} - \mathbf{v}^0)^2}{M} = \frac{\sum_p ((\mathbf{v}^{t^*}T - \mathbf{v}^0 T)/T)^2}{M} \tag{6}$$

$$= \frac{\sum_p ((\mathbf{x}^T - \mathbf{x}^0) - \mathbf{v}^0 T)^2}{MT^2} = \frac{\sum_p (\int_0^T (\mathbf{v}(t) - \mathbf{v}^0 T)dt)^2}{MT^2}, \tag{7}$$

where $M \gg N$ is the number of all samples. The term $(\mathbf{v}(t) - \mathbf{v}^0 T)$ is a typical (degree 0) polynomial interpolation error and can be defined as:

$$\epsilon_{\mathrm{IP}(0)}(t) = \mathbf{v}(t) - \mathbf{v}^0 T = \mathbf{v}(t) - P_k(t)\big|k = 0 \tag{8}$$

$$= \frac{\mathbf{v}^{(k+1)}(\xi)}{(k+1)!}(t - t_0)(t - t_1)...(t - t_k)\big|k = 0 = \mathbf{v}(\xi)t, \tag{9}$$

where $0 \leq \xi \leq T$, and the corresponding integration error is:

$$\epsilon_{\mathrm{NC}(0)} = \int_0^T \epsilon_{\mathrm{IP}(0)}(t)dt = \mathbf{v}(\xi)\int_0^T tdt = \frac{1}{2}\mathbf{v}(\xi)T^2 = \mathcal{O}(T^2). \tag{10}$$

Therefore, we obtain the final variance:

$$\sigma_{NC(0)}^2 = \frac{M\mathcal{O}(T^4)}{MT^2} = \mathcal{O}(T^2), \tag{11}$$

concluding the proof. □

### 3.3 NC (k)

In comparison with the 0 degree polynomial approximation NC(0), the high order NC($k$) is more accurate and yields only a little extra computational cost.

Suppose that we now have $K + 1$ (usually $K \leq 8$ due to the catastrophic Runge's phenomenon [42]) points $\mathbf{x}_i^0, \mathbf{x}_i^1, ..., \mathbf{x}_i^K$ equally spaced on time interval $[0, T]$, then the prediction for the future coordinate $\hat{\mathbf{x}}_i^T$ in Equation (4) can be re-written as:

$$\hat{\mathbf{x}}_i^T = \mathbf{x}_i^0 + \frac{T}{K} \sum_{k=0}^{K} w^k \hat{\mathbf{v}}_i^k, \tag{12}$$

where $\{w^k\}$ is the coefficients of Newton-Cotes formulas and $\hat{\mathbf{v}}_i^k$ denotes the predicted velocity at time $t^k$. To obtain $\hat{\mathbf{v}}_i^k, k \geq 1$, we simply regard EGNN as a recurrent model [43, 44] and re-input the last output velocity and coordinate. Some popular sequence models like LSTM [45] or Transformer [13] may be also leveraged, which we leave to future work. Then, the equation of updating the predicted velocity $\hat{\mathbf{v}}_i^k$ at $t^k$ can be written as:

$$\hat{\mathbf{v}}_i^k = \phi_v(\mathbf{h}_i)\mathbf{v}_i^{k-1} + \frac{\sum_{j \neq i} (\mathbf{x}_i^{k-1} - \mathbf{x}_j^{k-1})\mathbf{m}_{ij}}{N - 1}, \tag{13}$$

where $\mathbf{v}_i^{k-1}, \mathbf{x}_i^{k-1}$ denote the velocity and coordinate at $t^{k-1}$, respectively. Note that, Equation (13) is different from the form used in a multi-layer EGNN. The latter always uses the same input velocity and coordinate as constant features in different layers. By contrast, we first predict the next velocity and coordinate and then use them as input to get the new trainable velocity and coordinate. This process is more like training a language model [13, 43, 46, 45, 47].

From the interpolation theorem [26, 48], there exists a unique polynomial $\sum_{k=0}^{K} C^k t^{(k)}$ of degree at most $K$ that interpolates the $K + 1$ points, where $C^k$ is the coefficients of the polynomial. To avoid ambiguity, we here use $t^{(k)}$ to denote $t$ raised to the power of $k$, whereas $t^k$ to denote the $k$-th time point. $\sum_{k=0}^{K} C^k t^{(k)}$ thus can be constructed by the following equation:

$$\sum_{k}^{K} C^k t^{(k)} = [\mathbf{v}(t^0)] + [\mathbf{v}(t^0), \mathbf{v}(t^1)](t - t^0) + ... + [\mathbf{v}(t^0), ..., \mathbf{v}(t^K)](t - t^0)(t - t^1)...(t - t^K), \tag{14}$$

where $[\cdot]$ denotes the divided difference. In our case, the $K + 1$ points are equally spaced over the interval $[0, T]$. According to Newton-Cotes formulas, we will have:

$$\int_0^T \mathbf{v}(t)dt \approx \int_0^T \sum_{k=0}^{K} C_k t^{(k)} dt = \frac{T}{K} \sum_{k=0}^{K} w^k \mathbf{v}^k. \tag{15}$$

Particularly, $\{w^k\}$ are Newton-Cotes coefficients that are irrelevant to $\mathbf{v}(t)$. With additional observable points, the estimation for the integration can be much more accurate. The objective thus can be written as follows:

$$\underset{\{\hat{\mathbf{v}}^k\}}{\operatorname{argmin}} \sum_p \mathbf{x}^T - \hat{\mathbf{x}}^T = \sum_p \mathbf{x}^T - \mathbf{x}^0 - \frac{T}{K} \sum_{k=0}^{K} w^k \hat{\mathbf{v}}^k. \tag{16}$$

**Proposition 3.3** (Variance of $\sigma_{NC(k)}^2$). $\forall k \in \{0, 1, ...\}, \epsilon_{NC(k)} \geq \epsilon_{NC(k+1)}$, and consequently:

$$\sigma_{NC(0)}^2 \geq \sigma_{NC(1)}^2 \geq ... \geq \sigma_{NC(k)}^2. \tag{17}$$

*Proof.* Please see Appendix B.2 for details. Briefly, we only need to prove that $\epsilon_{NC(0)} \geq \epsilon_{NC(1)}$, where NC (1) is associated with Trapezoidal rule. $\square$

**Proposition 3.4** (Equivariance of NC). *NC possesses the equivariance property if the backbone model $\mathcal{M}$ (e.g., EGNN) in NC possesses this property.*

*Proof.* Please see Appendix B.3. $\square$

---

**Algorithm 1** Newton-Cotes Graph Neural Network

---
1: **Input:** the dataset $\mathcal{D}$, the backbone model $\mathcal{M}$, number of steps $k$ for NC;
2: **repeat**
3:    **for each** batched data in the training set $(\{\boldsymbol{X}^0, ..., \boldsymbol{X}^k\}, \{\boldsymbol{V}^0, ..., \boldsymbol{V}^k\})$ **do**
4:       $\hat{\boldsymbol{X}}^0 \leftarrow \boldsymbol{X}^0, \quad \hat{\boldsymbol{V}}^0 \leftarrow \boldsymbol{V}^0$;
5:       **for** $i := 1$ **to** $k$ **do**
6:          $\hat{\boldsymbol{X}}^i, \hat{\boldsymbol{V}}^i \leftarrow \mathcal{M}(\hat{\boldsymbol{X}}^{i-1}, \hat{\boldsymbol{V}}^{i-1})$;
7:       **end for**
8:       Compute the final prediction $\hat{\boldsymbol{X}}^k$ by Equation 12;
9:       Compute the main prediction loss $\mathcal{L}_{main}$ according to Equation (16);
10:      Compute the velocity regularization loss $\mathcal{L}_r$ according to Equation (18);
11:      Minimize $\mathcal{L}_{main}, \mathcal{L}_r$;
12:    **end for**
13: **until** the loss on the validation set converges.

---

### 3.4 NC (k) and NC$^+$ (k)

In the previous formulation, in addition to the initial and terminal points, we also need the other $K - 1$ points for supervision which yields a regularization loss:

$$\mathcal{L}_r = \sum_p \sum_{k=0}^{K} \|\mathbf{v}^k - \hat{\mathbf{v}}^k\|. \tag{18}$$

We denote NC (k) with the above loss by NC$^+$ (k). Minimizing $\mathcal{L}_r$ is equivalent to minimizing the difference between the true velocity and predicted velocity at each time point $t^k$ for each particle. $\|\cdot\|$ can be arbitrary distance measure, e.g., L2 distance. Although the $K - 1$ points of data can be easily accessed in practice, we argue that a loose version (without $\mathcal{L}_r$) NC (k) can also learn promising estimation of the integration $\int_0^T \mathbf{v}(t)dt$. Specifically, we still use $K + 1$ points $\{\hat{\mathbf{v}}^k\}$ to calculate the integration over $[0, T]$, except that the intermediate $K - 1$ points are unbounded. The model itself needs to determine the proper values of $\{\hat{\mathbf{v}}^k\}$ to optimize the same objective defined in Equation 16. In Section 4.5, we design an experiment to investigate the difference between the true velocities $\{\mathbf{v}^k\}$ and the predicted velocities $\{\hat{\mathbf{v}}^k\}$.

### 3.5 Computational Complexity and Limitations

In comparison with EGNN, NC (k) does not involve additional neural layers or parameters. Intuitively, we recurrently use the last output of EGNN to produce new predicted velocities at next time point, the corresponding computational cost should be $K$ times that of the original EGNN. However, our experiment in Section 4.4 shows that the training time did not actually linearly increase w.r.t. $K$. One reason may be the gradient in back-propagation is still computed only once rather than $K$ times.

We present an implementation of NC$^+$ in Algorithm 1. We first initialize all variables in the model and then recurrently input the last output to the backbone model to collect a series of predicted velocities. We then calculate the final predicted integration and plus the initial coordinate as the final output (i.e., Equation (12)). Finally we compute the losses and update the model by back-propagation.

## 4 Experiment

We conducted experiments on three different tasks towards reasoning system dynamics. The source code and datasets are available at https://github.com/zjukg/NCGNN.

### 4.1 Settings

We selected three state-of-the-art methods as our backbone models: RF [17], EGNN [18], and GMN [19]. To ensure a fair comparison, we followed their best parameter settings (e.g., hidden size and the number of layers) to construct NC. In other words, the main settings of NC and the baselines

Table 1: Prediction error ($\times 10^{-2}$) on N-body dataset. The header of each column "$p, s, h$" denotes the scenario with $p$ isolated particles, $s$ sticks and $h$ hinges. Results averaged across 3 runs.

| | |Train| = 500 | | | | | |Train| = 1500 | | | | |
| | 1,2,0 | 2,0,1 | 3,2,1 | 0,10,0 | 5,3,3 | 1,2,0 | 2,0,1 | 3,2,1 | 0,10,0 | 5,3,3 |
|---|---|---|---|---|---|---|---|---|---|---|
| Linear | $8.23_{\pm0.00}$ | $7.55_{\pm0.00}$ | $9.76_{\pm0.00}$ | $11.36_{\pm0.00}$ | $11.62_{\pm0.00}$ | $8.22_{\pm0.00}$ | $7.55_{\pm0.00}$ | $9.76_{\pm0.00}$ | $11.36_{\pm0.00}$ | $11.62_{\pm0.00}$ |
| GNN [9] | $5.33_{\pm0.07}$ | $5.01_{\pm0.08}$ | $7.58_{\pm0.08}$ | $9.83_{\pm0.04}$ | $9.77_{\pm0.02}$ | $3.61_{\pm0.13}$ | $3.23_{\pm0.07}$ | $4.73_{\pm0.11}$ | $7.97_{\pm0.44}$ | $7.91_{\pm0.31}$ |
| TFN [39] | $11.54_{\pm0.38}$ | $9.87_{\pm0.27}$ | $11.66_{\pm0.08}$ | $13.43_{\pm0.31}$ | $12.23_{\pm0.12}$ | $5.86_{\pm0.35}$ | $4.97_{\pm0.23}$ | $8.51_{\pm0.14}$ | $11.21_{\pm0.21}$ | $10.75_{\pm0.08}$ |
| SE(3)-Tr. [29] | $5.54_{\pm0.06}$ | $5.14_{\pm0.03}$ | $8.95_{\pm0.04}$ | $11.42_{\pm0.01}$ | $11.59_{\pm0.01}$ | $5.02_{\pm0.03}$ | $4.68_{\pm0.05}$ | $8.39_{\pm0.02}$ | $10.82_{\pm0.03}$ | $10.85_{\pm0.02}$ |
| RF [17] | $3.50_{\pm0.17}$ | $3.07_{\pm0.24}$ | $5.25_{\pm0.44}$ | $7.59_{\pm0.25}$ | $7.73_{\pm0.39}$ | $2.97_{\pm0.15}$ | $2.19_{\pm0.11}$ | $3.80_{\pm0.25}$ | $5.71_{\pm0.31}$ | $5.66_{\pm0.27}$ |
| NC (RF) | $\mathbf{2.84}_{\pm0.01}$ | $2.35_{\pm0.04}$ | $4.32_{\pm0.08}$ | $\mathbf{6.67}_{\pm0.26}$ | $7.14_{\pm0.25}$ | $2.91_{\pm0.11}$ | $\mathbf{1.76}_{\pm0.01}$ | $3.23_{\pm0.01}$ | $5.09_{\pm0.05}$ | $5.34_{\pm0.03}$ |
| NC$^+$ (RF) | $2.92_{\pm0.10}$ | $\mathbf{2.34}_{\pm0.03}$ | $\mathbf{4.28}_{\pm0.05}$ | $7.02_{\pm0.14}$ | $\mathbf{7.06}_{\pm0.33}$ | $\mathbf{2.87}_{\pm0.05}$ | $1.78_{\pm0.01}$ | $3.24_{\pm0.02}$ | $\mathbf{5.06}_{\pm0.06}$ | $\mathbf{5.23}_{\pm0.29}$ |
| EGNN [18] | $2.81_{\pm0.12}$ | $2.27_{\pm0.04}$ | $4.67_{\pm0.07}$ | $4.75_{\pm0.05}$ | $4.59_{\pm0.07}$ | $2.59_{\pm0.10}$ | $1.86_{\pm0.02}$ | $2.54_{\pm0.01}$ | $2.79_{\pm0.04}$ | $3.25_{\pm0.07}$ |
| NC (EGNN) | $2.41_{\pm0.03}$ | $2.18_{\pm0.02}$ | $3.53_{\pm0.01}$ | $4.26_{\pm0.03}$ | $4.13_{\pm0.03}$ | $\mathbf{2.23}_{\pm0.13}$ | $1.91_{\pm0.03}$ | $\mathbf{2.14}_{\pm0.03}$ | $2.36_{\pm0.05}$ | $\mathbf{2.86}_{\pm0.03}$ |
| NC$^+$ (EGNN) | $\mathbf{2.25}_{\pm0.01}$ | $\mathbf{2.07}_{\pm0.06}$ | $\mathbf{2.01}_{\pm0.02}$ | $\mathbf{3.54}_{\pm0.06}$ | $\mathbf{3.96}_{\pm0.04}$ | $2.32_{\pm0.05}$ | $\mathbf{1.86}_{\pm0.13}$ | $2.39_{\pm0.01}$ | $\mathbf{2.35}_{\pm0.07}$ | $2.99_{\pm0.02}$ |
| GMN [19] | $1.84_{\pm0.02}$ | $2.02_{\pm0.02}$ | $2.48_{\pm0.04}$ | $2.92_{\pm0.04}$ | $4.08_{\pm0.03}$ | $1.68_{\pm0.04}$ | $1.47_{\pm0.03}$ | $2.10_{\pm0.04}$ | $2.32_{\pm0.02}$ | $2.86_{\pm0.01}$ |
| NC (GMN) | $\mathbf{1.55}_{\pm0.07}$ | $1.58_{\pm0.02}$ | $2.07_{\pm0.03}$ | $2.73_{\pm0.02}$ | $2.99_{\pm0.03}$ | $1.59_{\pm0.03}$ | $1.18_{\pm0.05}$ | $1.47_{\pm0.01}$ | $\mathbf{1.66}_{\pm0.01}$ | $1.93_{\pm0.03}$ |
| NC$^+$ (GMN) | $1.57_{\pm0.02}$ | $\mathbf{1.43}_{\pm0.02}$ | $\mathbf{2.03}_{\pm0.04}$ | $\mathbf{2.57}_{\pm0.04}$ | $\mathbf{2.72}_{\pm0.03}$ | $\mathbf{1.49}_{\pm0.02}$ | $\mathbf{1.17}_{\pm0.04}$ | $\mathbf{1.44}_{\pm0.01}$ | $1.70_{\pm0.06}$ | $\mathbf{1.91}_{\pm0.02}$ |

Table 2: Prediction error ($\times 10^{-2}$) on MD17 dataset. Results averaged across 3 runs.

| | Aspirin | | | Benzene | | | Ethanol | | | Malonaldehyde | | |
| | RF | EGNN | GMN | RF | EGNN | GMN | RF | EGNN | GMN | RF | EGNN | GMN |
|---|---|---|---|---|---|---|---|---|---|---|---|---|
| Raw | $10.94_{\pm0.01}$ | $14.41_{\pm0.15}$ | $10.14_{\pm0.03}$ | $103.72_{\pm1.29}$ | $62.40_{\pm0.53}$ | $48.12_{\pm0.40}$ | $4.64_{\pm0.01}$ | $4.64_{\pm0.01}$ | $\mathbf{4.83}_{\pm0.01}$ | $13.93_{\pm0.03}$ | $13.64_{\pm0.01}$ | $13.11_{\pm0.03}$ |
| NC | $\mathbf{10.87}_{\pm0.01}$ | $9.63_{\pm0.01}$ | $9.50_{\pm0.06}$ | $63.22_{\pm0.01}$ | $55.05_{\pm1.60}$ | $41.62_{\pm1.16}$ | $\mathbf{4.62}_{\pm0.01}$ | $4.63_{\pm0.01}$ | $\mathbf{4.83}_{\pm0.01}$ | $\mathbf{12.93}_{\pm0.03}$ | $12.82_{\pm0.01}$ | $12.97_{\pm0.01}$ |
| NC$^+$ | $\mathbf{10.87}_{\pm0.01}$ | $\mathbf{9.60}_{\pm0.02}$ | $\mathbf{9.40}_{\pm0.09}$ | $63.04_{\pm0.05}$ | $\mathbf{54.76}_{\pm0.74}$ | $\mathbf{41.26}_{\pm0.15}$ | $\mathbf{4.62}_{\pm0.01}$ | $\mathbf{4.62}_{\pm0.01}$ | $\mathbf{4.83}_{\pm0.01}$ | $\mathbf{12.93}_{\pm0.03}$ | $\mathbf{12.81}_{\pm0.02}$ | $\mathbf{12.92}_{\pm0.01}$ |

| | Naphthalene | | | Salicylic | | | Toluene | | | Uracil | | |
| | RF | EGNN | GMN | RF | EGNN | GMN | RF | EGNN | GMN | RF | EGNN | GMN |
|---|---|---|---|---|---|---|---|---|---|---|---|---|
| Raw | $0.50_{\pm0.01}$ | $0.47_{\pm0.02}$ | $0.40_{\pm0.01}$ | $1.23_{\pm0.01}$ | $1.02_{\pm0.02}$ | $0.91_{\pm0.01}$ | $10.93_{\pm0.04}$ | $11.78_{\pm0.07}$ | $10.22_{\pm0.08}$ | $0.64_{\pm0.01}$ | $0.64_{\pm0.01}$ | $0.59_{\pm0.01}$ |
| NC | $\mathbf{0.39}_{\pm0.01}$ | $\mathbf{0.37}_{\pm0.01}$ | $0.38_{\pm0.01}$ | $\mathbf{1.09}_{\pm0.01}$ | $\mathbf{0.84}_{\pm0.01}$ | $\mathbf{0.86}_{\pm0.01}$ | $\mathbf{10.85}_{\pm0.01}$ | $10.64_{\pm0.11}$ | $10.17_{\pm0.04}$ | $\mathbf{0.60}_{\pm0.01}$ | $\mathbf{0.57}_{\pm0.01}$ | $\mathbf{0.56}_{\pm0.01}$ |
| NC$^+$ | $\mathbf{0.39}_{\pm0.01}$ | $\mathbf{0.37}_{\pm0.01}$ | $\mathbf{0.37}_{\pm0.02}$ | $\mathbf{1.09}_{\pm0.01}$ | $\mathbf{0.84}_{\pm0.01}$ | $\mathbf{0.86}_{\pm0.01}$ | $\mathbf{10.85}_{\pm0.01}$ | $\mathbf{10.52}_{\pm0.01}$ | $\mathbf{10.14}_{\pm0.01}$ | $\mathbf{0.60}_{\pm0.01}$ | $\mathbf{0.57}_{\pm0.01}$ | $\mathbf{0.56}_{\pm0.02}$ |

were identical. The results reported in the main experiments were based on a model of NC (2) for a balance of training cost and performance.

We reported the performance of NC w/ additional points (denoted by $\mathbf{NC^+}$) and w/o additional points (the relaxed version, denoted by **NC**).

## 4.2 Datasets

We leveraged three datasets towards different aspects of reasoning dynamics to exhaust the performance of NC. We used the N-body simulation benchmark proposed in [19]. Compared with the previous ones, this benchmark has more different settings, e.g., number of particles and training data. Furthermore, it also considers some physical constraints (e.g., hinges and sticks) widely existing in the real world. For molecular dynamics, We used MD17 [28] that consists of the molecular dynamics simulations of eight different molecules as a scenario. We followed [19] to construct the training/testing sets. We also considered reasoning human motion. Follow [19], the dataset was constructed based on 23 trials in CMU Motion Capture Database [27].

As NC$^+$ requires additional supervised data, we accordingly extracted more intermediate data points between the input and target from each dataset. It is worth noting that this operation was tractable as the sampling frequency of the datasets was much higher than our requirement.

## 4.3 Main Results

We reported the main experimental results w.r.t. three datasets in Tables 1, 2, and 3, respectively. Overall, NC$^+$ significantly improved the performance of all baselines on almost all datasets and across all settings, which empirically demonstrates the generality as well as the effectiveness of the proposed method. Remarkably, NC (without extra data) achieved slightly low results but still had great advantages over performance in comparison with three baselines.

Specifically, The average improvement on N-body datasets was most significant, where we observe a near 20% decrease in prediction error for all three baseline methods. For example, NC (RF) and NC$^+$

Table 3: Prediction error ($\times 10^{-2}$) on motion capture. Results averaged across 3 runs.

|      | GNN           | TFN           | SE(3)-Tr.    | RF             | EGNN          | GMN           |
|------|---------------|---------------|--------------|----------------|---------------|---------------|
| Raw  | $67.3_{\pm 1.1}$ | $66.9_{\pm 2.7}$ | $60.9_{\pm 0.9}$ | $197.0_{\pm 1.0}$ | $59.1_{\pm 2.1}$ | $43.9_{\pm 1.1}$ |
| NC   | N/A           | N/A           | N/A          | $164.8_{\pm 1.7}$ | $55.8_{\pm 6.1}$ | $30.0_{\pm 1.4}$ |
| NC$^+$ | N/A         | N/A           | N/A          | $\mathbf{162.6_{\pm 0.8}}$ | $\mathbf{51.3_{\pm 4.0}}$ | $\mathbf{29.6_{\pm 0.8}}$ |

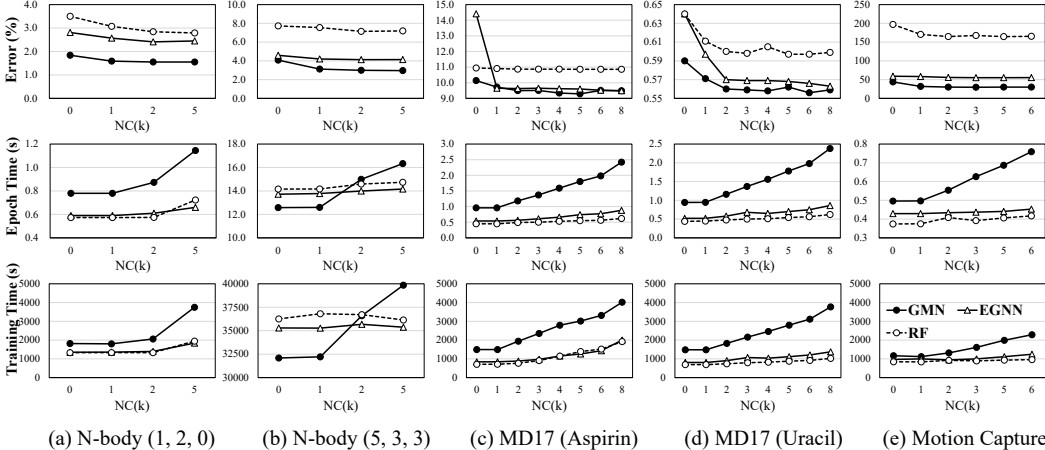

| (a) N-body (1, 2, 0) | (b) N-body (5, 3, 3) | (c) MD17 (Aspirin) | (d) MD17 (Uracil) | (e) Motion Capture |

Figure 2: A detailed comparison of NCGNN without extra data on 5 datasets, w.r.t. $k$, average of 3 runs, conducted on a V100. The lines with dot, circle, and triangle denote the results of GMN, EGNN, and RF, respectively. The first, second, and third rows are the results of prediction error, training time of one epoch, and total training time, respectively.

(RF) greatly outperformed the original RF and even had better results than the original EGNN under some settings (e.g., 3,2,1). This phenomenon also happened on the molecular dynamics datasets, where we find that NC$^+$ (EGNN) not only defeated its original version, but also the original GMN and even NC$^+$ (GMN) on many settings.

## 4.4 Impact of estimation step $k$

We conducted experiments to explore how the performance changes with respect to the estimation step $k$. The experimental results are shown in Figure 2.

For all three baselines, the performance improved rapidly as the growth of step $k$ from 0 to 2, but this advantage gradually vanished as we set larger $k$. Interestingly, the curve of training time showed an inverse trend, where we find that the total training time of NC (5) with GMN was almost two times slower than the NC (1) version. Therefore, we set $k = 2$ in previous experiments to get the best balance between performance and training cost.

By comparing the prediction errors of different baselines, we find that GMN that considers the physical constraints (e.g., sticks and hinges) usually had better performance than the other two methods, but the gap significantly narrowed with the increase of step $k$. For example, on MD17 and motion datasets, EGNN obtained similar prediction errors for $k \geq 2$ while demanding less training time. This phenomenon demonstrates that predicting the integration with multiple estimations lowers the requirement for model capacity.

Overall, learning with NC significantly reduced the prediction errors of different models. The additional training time was also acceptable in most cases.

## 4.5 A Comparison between NC and NC$^+$

The only difference between NC and NC$^+$ was the use of the regularization loss in NC$^+$ to learn the intermediate velocities. Therefore, we designed experiments to compare the intermediate results and the final predictions of NC and NC$^+$ with GMN as the backbone model.

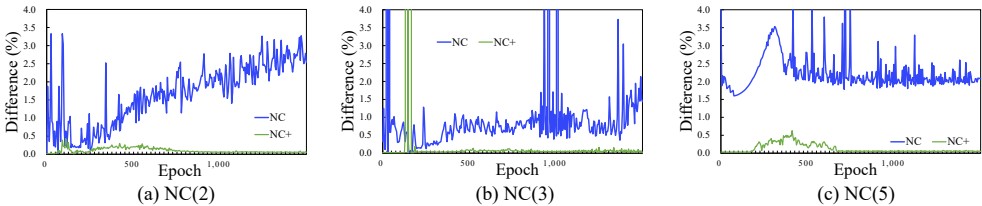

(a) NC(2)  (b) NC(3)  (c) NC(5)

Figure 3: The prediction errors of intermediate velocities on valid set, w.r.t. training epoch. The blue and green lines denote the prediction errors of NC and NC$^+$, respectively.

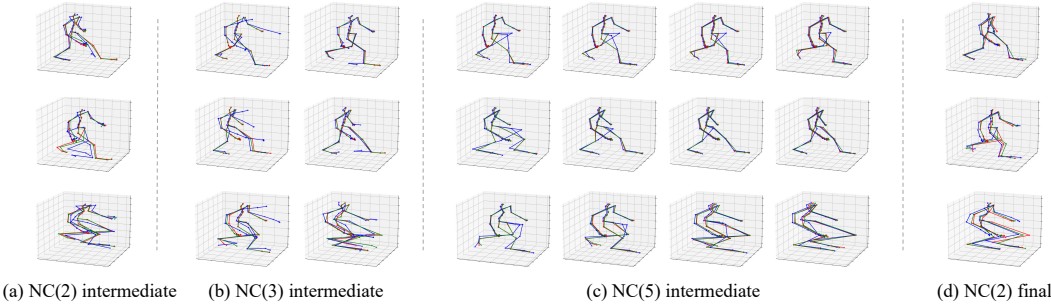

(a) NC(2) intermediate  (b) NC(3) intermediate  (c) NC(5) intermediate  (d) NC(2) final

Figure 4: Visualization of the intermediate velocities w.r.t. $k$. The red, blue, and green lines denote the target, prediction of NC, and prediction of NC$^+$, respectively.

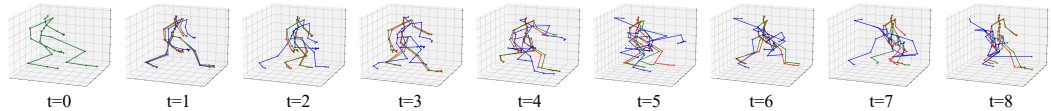

t=0   t=1   t=2   t=3   t=4   t=5   t=6   t=7   t=8

Figure 5: Consecutive predictions on the Motion dataset. The red, blue, and green lines denote the target, the prediction of NC (0), and the prediction of NC$^+$ (2), respectively. The interval between two figures is identical to the setting in the main experiment.

We first tracked the average intermediate velocity prediction errors on valid datasets in terms of training epochs. The results are shown in Figure 3, from which we observe that the prediction errors were not huge for both two methods, especially NC$^+$ obtained highly accurate prediction for the intermediate velocities with different $k$. For NC, we actually find that it implicitly learned to model the intermediate velocities more or less, even though its prediction errors gradually improved or dramatically fluctuated during training. This observation empirically demonstrates the effectiveness of the intermediate velocities in predicting the final state of the system. We also reported the results on MD17 and N-body datasets in Appendix C.

We then visualized the intermediate velocities and the final predictions in Figure 4. The intermediate velocities were visualized by adding the velocity with the corresponding coordinate at the intermediate time point. From Figure 4, we can find that NC$^+$ (green ones) learned very accurate predictions with only a few non-overlapped nodes to the target nodes (red ones). For the loose version NC, it learned good predictions for the backbone part, but failed on the limbs (especially the legs). The motion of the main backbone was usually stable, while that of the limbs involved swing and twisting. We also conducted experiments on the other two types of datasets, please see Figure 8 in Appendix C.

### 4.6 On the Time Evolution of Dynamic Systems

In previous experiments, we show that NC$^+$ was capable of learning highly accurate predictions of intermediate velocities. This inspired us to realize the full potential of NC$^+$ by producing a series of predictions given only the initial state as input. For each time point (including the intermediate time points), we used the most recent $k + 1$ points of averaged data as input to predict the next velocity and

Table 4: Multi-step prediction results ($\times 10^{-2}$) on the N-body and MD17 datasets. ADE and FDE denote average displacement error and final displacement error, respectively. The results of four baseline methods are from [49].

| | N-body (Springs) | | MD17 (Aspirin) | | MD17 (Benzene) | | MD17 (Ethanol) | | MD17 (Malonaldehyde) | |
| --- | --- | --- | --- | --- | --- | --- | --- | --- | --- | --- |
| | ADE | Accuracy | ADE | FDE | ADE | FDE | ADE | FDE | ADE | FDE |
| LSTM [45] | - | 53.5 | 17.59 | 24.79 | 6.06 | 9.46 | 7.73 | 9.88 | 15.14 | 22.90 |
| NRI [50] | - | 93.0 | 12.60 | 18.50 | 1.89 | 2.58 | 6.69 | 8.78 | 12.79 | 19.86 |
| GroupNet [51] | - | - | 10.62 | 14.00 | 2.02 | 2.95 | 6.00 | 7.88 | 7.99 | 12.49 |
| EqMotion [49] | 16.8 | 97.6 | 5.95 | 8.38 | 1.18 | 1.73 | 5.05 | 7.02 | 5.85 | 9.02 |
| NC (EqMotion) | **16.2** | **98.9** | **4.74** | **6.76** | **1.15** | **1.63** | **5.02** | **6.87** | **5.19** | **8.07** |

coordinate. This strategy produced much more stable predictions than the baseline NC (0) – directly using the last output as input to predict the next state.

The results are shown in Figure 5, from which we can find that both the baseline (blue lines) and NC$^+$ (green lines) had good predictions at the initial steps. However, due to the increasing acculturated errors, the baseline soon collapsed and only the main backbone was roughly fit to the target. By contrast, NC$^+$ was still able to produce promising results. The deviations mainly concentrated on the limbs. Therefore, we argue that NC$^+$ can produce not only the highly accurate one-step prediction, but also the relatively reliable consecutive predictions. We also uploaded the .gif files of examples on all three datasets in Supplementary Materials, which demonstrated the consistent conclusion.

We also compared our method with the EGNN methods for multi-step prediction [49–52]. Specifically, multi-step prediction takes a sequence of states with fixed interval as input and predicts a sequence of future states with same length and interval. As recent multi-step prediction methods are still based on sequence or encoder-decoder architectures, it would be interesting to examine the applicability of NC to them. To this end, we adapted NC ($k$=2) to the source code of the best-performing method EqMotion [49] and used identical settings for the hyper-parameters.

The experimental results are shown in Table 4, where we can observe that NC (EqMotion) outperformed the base model EqMotion on all datasets and across all metrics. It is worth noting that the archiecture and parameter-setting of the EGNN module were identical to EqMotion and NC (EqMotion), which clearly demonstrates the effectiveness of the proposed method. We have also released the source code of NC (EqMotion) in our GitHub repository.

## 4.7 Ablation Study

We conducted ablation studies to investigate the effectiveness of the proposed Newton-Cotes integration. We implemented two alternative methods: NC (coeffs= 1), where we set all the coefficients for summing the predicted velocities as 1; and depth-varying ODE, where we employed common neural ODE solvers from [40, 41] to

Table 5: Ablation study results ($\times 10^{-2}$).

| | N-body (1,2,0) | MD17 (Aspirin) | Motion |
| --- | --- | --- | --- |
| GMN | 1.84 | 10.14 | 43.9 |
| NC (coeffs= 1) | 1.78 | 9.64 | 48.4 |
| depth-varying ODE | 1.82 | 10.08 | 42.2 |
| NC | 1.55 | 9.50 | 30.0 |

address our problem. We used an identical GMN as the backend EGNN model and conducted a simple hyper-parameter search for the learning rate and ODE solver.

The results are presented in Table 5. The two alternative methods methods generally performed better than the baseline GMN. The performance of depth-varying GDE was comparable to that of NC (coeffs= 1), but both were lower than that of the original NC. This observation further supports the effectiveness of our method.

## 5 Conclusion

In this paper, we propose NC to tackle the problem of reasoning system dynamics. We show that the existing state-of-the-art methods can be derived to the basic form of NC, and theoretically/empirically prove the effectiveness of high order NC. Furthermore, with a few additional data provided, NC$^+$ is capable of producing promising consecutive predictions. In future, we plan to study how to improve the ability of NC$^+$ on the consecutive prediction task.

## Acknowledgement

We would like to thank all anonymous reviewers for their insightful and invaluable comments. This work is funded by NSFCU19B2027/NSFC91846204/NSFC62302433, National Key R&D Program of China (Funding No.SQ2018YFC000004), Zhejiang Provincial Natural Science Foundation of China (No.LGG22F030011) and sponsored by CCF-Tencent Open Fund (CCF-Tencent RAGR20230122).

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

# A  The Derivations of NC (0) for Other EGNN Models

In this section, we first illustrate how the other two baseline models (i.e. RF [17] and GMN [19]) can be derived to NC (0), and then discuss multi-layer EGNN.

## A.1  RF

The overall equivariant convolution process of RF is similar to that of EGNN. The main difference is that RF does not use the node feature. Instead, it leverages the L2 norm of the velocity as an additional feature to update the predicted velocity:

$$\hat{\mathbf{v}}_i^0 = \text{Norm}(\mathbf{v}_i^0)\big(\phi_v(\mathbf{h}_i)\mathbf{v}_i^0 + \frac{1}{N-1}\sum_{j \neq i}(\mathbf{x}_i^0 - \mathbf{x}_j^0)\mathbf{m}_{ij}^0\big), \tag{19}$$

$$\hat{\mathbf{x}}_i^T = \mathbf{x}_i^0 + \hat{\mathbf{v}}_i^0 T, \tag{20}$$

where $\text{Norm}(\mathbf{v}_i^0)$ denotes the L2 norm of the input velocity $\mathbf{v}_i^0$. Therefore, RF can be also viewed as a NC (0) model.

## A.2  GMN

The main difference between GMN and EGNN is that GMN detects the sub-structures in the system and process the particles in each special sub-structure independently. Specifically, GMN re-formulates the original EGNN (i.e., Equations (2-4)) as follows:

$$\mathbf{m}_{ij}^0 = \phi_e(\mathbf{h}_i, \mathbf{h}_j, ||\mathbf{x}_i^0 - \mathbf{x}_j^0||^2, e_{ij}), \tag{21}$$

$$\hat{\mathbf{v}}_k^0 = \phi_v(\sum_{i \in \mathcal{S}_k} \mathbf{h}_i)\mathbf{v}_k^0 + \sum_{i \in \mathcal{S}_k} \phi_k(\mathbf{x}_i^0, \mathbf{m}_{ij}^0), \tag{22}$$

$$\hat{\mathbf{v}}_i^0 = \text{FK}(\hat{\mathbf{v}}_k^0), \tag{23}$$

$$\hat{\mathbf{x}}_i^T = \mathbf{x}_i^0 + \hat{\mathbf{v}}_i^0 T, \tag{24}$$

where $\hat{\mathbf{v}}_k^0$ is the predicted velocity of the sub-structure. GMN then uses it to calculate the velocity of each particle by a function FK which can be either learnable or based on the angles and relative positions in the sub-structure [19].

## A.3  Multi-layer EGNN

In current EGNN methods, the input coordinate $\mathbf{x}_i^0$ and velocity $\mathbf{v}_i^0$ are regarded as constant feature and used in different layers. For example, in the multi-layer version EGNN, $\mathbf{m}_{ij}^0$ will be formulated as follows:

$$\mathbf{m}_{ij}^{0,l} = \phi_e(\mathbf{h}_i^{l-1}, \mathbf{h}_j^{l-1}, ||\mathbf{x}_i^0 - \mathbf{x}_j^0||^2, e_{ij}), \tag{25}$$

where $\mathbf{m}_{ij}^{0,l}$ is $\mathbf{m}_{ij}^0$ in layer $l$ and $\mathbf{h}_i^{l-1}$ denotes the hidden feature in layer $l-1$. Evidently, only the hidden features are updated within different layers. The overall formulation of multi-layer layer is akin to the single-layer EGNN.

# B Proofs of Things

## B.1 Proof of Proposition 3.1

*Proof.* The objective of the existing methods for a single system can be defined as:

$$\underset{\hat{\mathbf{v}}^0}{\operatorname{argmin}} \sum_{p_i} (\mathbf{x}_i^T - \hat{\mathbf{x}}_i^T) \tag{26}$$

$$= \sum_{p_i} (\mathbf{x}_i^T - \mathbf{x}_i^0 - \hat{\mathbf{v}}_i^0 T) \tag{27}$$

$$= \sum_{p_i} T(\frac{\mathbf{x}_i^T - \mathbf{x}_i^0}{T} - \hat{\mathbf{v}}_i^0) \tag{28}$$

$$= T \sum_{p_i} \left( \mathbf{v}_i^{t^*} - (\phi_v(\mathbf{h}_i)\mathbf{v}_i^0 + \frac{\sum_{j \neq i} (\mathbf{x}_i^0 - \mathbf{x}_j^0)\mathbf{m}_{ij}^0}{N-1}) \right) \tag{29}$$

$$= T \sum_{p_i} \left( \mathbf{v}_i^{t^*} - (w^0 \mathbf{v}_i^0 + \mathbf{b}^0) \right), \tag{30}$$

where $w^0 \in \mathbb{R}^1$ and $\mathbf{b}^0 \in \mathbb{R}^3$ denote the learnable variables irrelevant to $\mathbf{v}_i^0$ and $t$, concluding the proof. $\qquad\square$

## B.2 Proof of Proposition 3.3

*Proof.* As the higher order cases ($k \geq 1$) have already been proved, we only need to show that $\epsilon_{\text{NC}(0)} \geq \epsilon_{\text{NC}(1)}$. The first order of Newton-Cotes formula NC (1) is also known as Trapezoidal rule, i.e.:

$$\int_0^T \mathbf{v}(t)dt \approx \frac{T}{2}(\mathbf{v}^0 + \mathbf{v}^T). \tag{31}$$

As aforementioned, the actual integration $\mathbf{x}^T - \mathbf{x}^0$ for different training examples is different, and we assume that it fluctuates around the base estimation $\frac{T}{2}(\mathbf{v}^0 + \mathbf{v}^T)$ and follows a normal distribution $\mathcal{N}_{NC(1)}$, where the variance $\sigma_{NC(1)}^2$ is positively correlated with the difficulty of optimizing the overall objective. The variance of $\mathcal{N}_{NC(1)}$ is:

$$\sigma_{NC(1)}^2 = \frac{\sum_p (\int_0^T (\mathbf{v}(t) - \sum_{k=0}^1 C^k t^{(k)})dt)^2}{NT^2}, \tag{32}$$

where the integration term is a general form of polynomial interpolation error. According to Equation 14, it can be derived to:

$$\int_0^T (\frac{(t-t^0)(t-t^1)\mathbf{v}''(\xi)}{2!})dt, \tag{33}$$

where $\mathbf{v}''$ denote the second derivative of $\mathbf{v}$. Let $s = \frac{t-t^0}{T}$, then $t = t^0 + sh$ and $dt = d(\mathbf{v}^0 + sh) = Tds$. the above equation can be re-written as:

$$T \int_0^1 \frac{s(s-1)T^2\mathbf{v}''(\xi)}{2}ds = -\frac{1}{12}T^3\mathbf{v}''(\xi) = \mathcal{O}(T^3). \tag{34}$$

Therefore, the final $\sigma_{NC(1)}^2$ is:

$$\sigma_{NC(1)}^2 = \frac{\sum_p (-\frac{1}{12}T^3\mathbf{v}''(\xi))^2}{NT^2} \tag{35}$$

$$= \mathcal{O}(T^4) \leq \mathcal{O}(T^2) = \sigma_{NC(0)}^2, \tag{36}$$

concluding the proof. $\qquad\square$

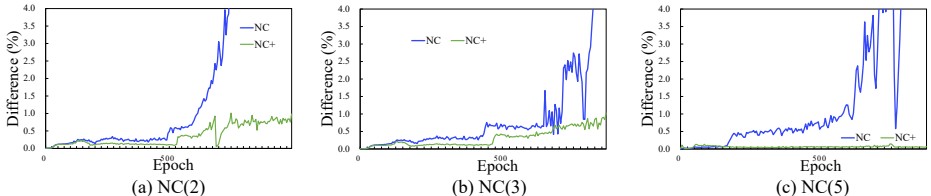

Figure 6: The prediction errors of intermediate velocities on MD17 dataset, w.r.t. training epoch. The blue and green lines denote the prediction errors of NC and NC$^+$, respectively.

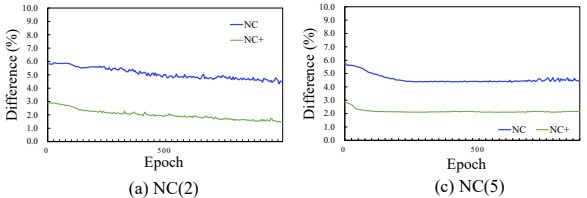

Figure 7: The prediction errors of intermediate velocities on N-body dataset, w.r.t. training epoch. The blue and green lines denote the prediction errors of NC and NC$^+$, respectively.

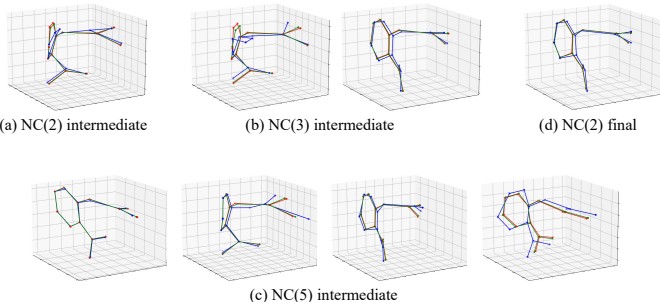

Figure 8: Visualization of the intermediate velocities w.r.t. $k$ of MD17 (Aspirin). The red, blue, and green lines denote the target, prediction of NC, and prediction of NC$^+$, respectively.

### B.3 Proof of Proposition 3.4

*Proof.* The GNN models possessing equivariance property are equivariant to the translation, rotation, and permutation of input. NC directly feeds the input into these backbone models and naturally possesses this property.

Formally, let $T_g : \mathbf{x} \rightarrow \mathbf{x}$ be a group of transformation operations. If the backbone model $\mathcal{M}$ is equivariant then we will have:

$$\mathcal{M}(T_g(\mathbf{x})) = S_g(\mathcal{M}(\mathbf{x})), \tag{37}$$

where $S_g$ is an equivalent transformation to $T_g$ on the output space. NC can be regarded as a weighted combination of the outputs of $\mathcal{M}$:

$$\sum_i w_i \mathcal{M}(T_g(\mathbf{x}_i))) = \sum_i w_i S_g(\mathcal{M}(\mathbf{x}_i)), \tag{38}$$

where the Newton-Cotes weights $w_i$ is constant and irrelevant to the input, the output, and the model $\mathcal{M}$ itself. Therefore, the above equation will always holds. $\square$

## C  Additional Experimental Results

The average intermediate velocity prediction errors on MD17 and Motion datasets are shown in Figure 6 and Figure 7, respectively. NC still learned the intermediate velocities we did not feed

Table 6: Hyper-parameter settings in the main experiments.

| Datasets | $k$ | velocity regularization | velocity regularization decay | parameter regularization | parameter regularization decay | loss criterion | input feature normalization | intermediate velocity normalization |
|---|---|---|---|---|---|---|---|---|
| N-body | 2 | 0.001 | 0.999 | 1.0 | 0.99 | MSE | False | True |
| MD17 | 2 | 0.1 | 0.999 | 1.0 | 0.95 | MSE | True | True |
| Motion | 2 | 0.01 | 0.999 | 1.0 | 0.95 | MSE | True | True |

| | # epoch | batch-size | # training examples | activation | # layers | learning rate | optimizer | clip gradient norm |
|---|---|---|---|---|---|---|---|---|
| N-body | 1,500 | 200 | 500 | ReLU | 4 | 0.0005 | Adam | 1.0 |
| MD17 | 1,000 | 100 | 500 | ReLU | 4 | 0.001 | Adam | 0.1 |
| Motion | 1,500 | 100 | 200 | ReLU | 4 | 0.0005 | Adam | 1.0 |

the intermediate into it. Particularly, the error on N-body dataset was small and stable, which may demonstrate the effectiveness of estimating intermediate velocities even without supervised data.

We also provide some visualized examples on MD17 dataset in Figure 8, from which we can observe the similar results compared with Figure 4 in the main paper.

# D Hyper-parameter Setting

We list the main hyper-parameter setting of NC with different EGNN models on different datasets in Table 6. We used the Adam optimizer [53] and adopted layer normalization [54] and ReLU activation [55]) for all settings. For a fair comparison, the parameter settings for the backbone EGNN models were identical to these in the existing implementation [19].

