# OpenReview forum: "Newton–Cotes Graph Neural Networks: On the Time Evolution of Dynamic Systems"
_NeurIPS.cc/2023/Conference — NeurIPS 2023 spotlight_

### Official Review · Reviewer_9Zk2 · 2023-07-02

**Soundness:** 2 fair
**Presentation:** 2 fair
**Contribution:** 1 poor
**Rating:** 4
**Confidence:** 4

**Summary:**

This paper proposes Newton-Cotes Graph Neural Networks (NC), which are a time-continuous extension of Equivariant Graph Neural Networks (EGNNs). The authors also conducted a mathematical analysis on the predictive variance of NC under the Gaussian assumption. Through numerical experiments, the authors have demonstrated that the NC-formulated EGNNs outperform the baseline EGNNs.

**Strengths:**

- The method is clear and can be generalized to other GNN classes beyond Equivariant GNNs.

**Weaknesses:**

**The proposed method is a special case of a larger GNN class, but there is no comparison provided with this GNN class in the submitted paper.**

Although the authors have demonstrated the effectiveness of the NC integrator in conjunction with GNN through numerical experiments, the paper fails to compare the proposed Newton-Cotes Graph Neural Networks (NC) with the most competitive baseline class, Graph Neural Ordinary Differential Equations (GDEs) [1].

It would be highly beneficial to compare the proposed method with the more well-established GDE models. In my opinion, the proposed NC can be considered a special case of GDE where the integrator is the Newton-Cotes method and the GNN corresponds to the proposed EGNN. Furthermore, based on the advancements in Neural ODE research, the limitation mentioned in section 3.5 could already be addressed using adjoint sensitivity methods.

[1] Graph Neural Ordinary Differential Equations

**Questions:**

- Regarding the selection of the numerical integrator, it would be beneficial to clarify the advantages of using NC over more well-known and commonly used integrators such as Runge-Kutta or Dormand–Prince.
- When it comes to training NC+, is it necessary to collect the K-1 intermediate points?

**Limitations:**

- The current manuscript lacks a comparison with the most competitive baseline methods, as well as a comprehensive review of relevant literature.

---

> ### Author Rebuttal · Authors · 2023-08-08
>
> Thank you very much for your insightful comments. We have taken them into account and provided clarifications below:
>
> **Weaknesses:**
>
> **1. The proposed method can be considered a special case of GDEs. It would be highly beneficial to compare the proposed method with GDEs.**
>
> Thanks very much for your insightful and detailed suggestion. We have added GDEs to our related works in the revision. GDEs propose a set of strategies to incorporate GNNs with neural ODEs:
>
> The depth-varying GDE, which can be regarded as “continuous” ResNet + GNNs. The GNN module acts as the derivative and can be stacked arbitrary times. In contrast, our paper uses the Newton-Cotes coefficients as a kind of prompt to encourage the model to fit the intermediate values and minimize the training loss. We have observed that NC outperforms the baselines significantly (Tables 1, 2, and 3), even without the additional data. We have also observed that NC implicitly learns the intermediate velocities in Figure 3. In our response to Reviewer Gu3W’s comment (Weakness 1), we implement a new variant of NC with coefficients set to 1. This variant is similar to the depth-varying GDE, but it performs worse than NC. To thoroughly investigate this issue, we have modified the source code of GDEs for our task. We use an identical GMN as the backend EGNN model and perform a simple hyper-parameter search for the learning rate and ODE solver. We report the best results as follows:
>
> | Method  |  N-body (1,2,0) | MD17 (Aspirin)  | Motion  |
> |---|---:|---:|---:|
> | GMN  | 1.84  | 10.14  | 43.9  |
> | NC (coefficients=1)  | 1.78  | 9.64  | 48.4  |
> | depth-varying GDE  | 1.82  | 10.08  | 42.2  |
> | NC   | 1.55  | 9.50  | 30.0  |
>
> The performance of depth-varying GDE is comparable to that of NC (coefficients=1), but both are lower than that of the original NC. All the methods generally perform better than the baseline GMN. This observation further supports the effectiveness of our method.
>
> Another related GDE model is the spatio-temporal GDE, but we are unable to adapt it to our task due to the unavailability of the source code. The spatio-temporal GDE, like many neural ODE methods, aims to build a continuous prediction and capture observations with time steps. However, it is not appropriate to consider our method as a special case of GDEs. To draw an analogy, it is acceptable to state that we use numerical integration algorithms to solve ODE problems, but it may sound peculiar to claim that we use ODE solvers to solve numerical integration. The numerical solvers (e.g., 4th order Runge-Kutta, rk4) in neural ODEs are originally designed for numerical integration. Regarding our method as a special case of GDEs is akin to treating numerical integration as an ODE problem and solving it using numerical algorithms. In contrast, it may be feasible to iteratively perform our method to solve problems in GDEs. Our work draws inspiration from observations on numerical integration algorithms, where the integrand is known only at certain points. The goal is to efficiently approximate the integral to the desired precision. Neural ODE methods repeatedly perform the numerical algorithms on small internals to obtain the numerical solution.
>
> Finally, we would like to emphasize that our contributions extend beyond proposing a new method. The theoretical analysis is equally important, as it provides a novel and comprehensive perspective on reasoning dynamics with neural models.
>
> **2. The limitation mentioned in section 3.5 could be addressed using adjoint sensitivity methods.**
>
> The adjoint sensitivity methods are generally used to reduce memory usage (at a cost of time). They cannot reduce the training time of our method.
>
> **Questions:**
>
> **1. It would be beneficial to clarify the advantages of using NC over other algorithms.**
>
> Thanks for your suggestion. We use NC because it aligns closely with our scenarios. The intervals between states are always constant in trajectories. This characteristic allows us to easily obtain the intermediate points that satisfy NC (but not other algorithms, e.g., Gaussian quadrature and Clenshaw–Curtis quadrature). We will include the above explanations in the revision.
>
> **2. When it comes to training NC+, is it necessary to collect the K-1 intermediate points?**
>
> We acknowledge the need for the intermediate points to train NC+ in a supervised fashion. Fortunately, the additional data can be readily obtained since the sampling frequency used in constructing the datasets is much higher than what we require.
>
> Once again, thank you for your valuable feedback. We hope our responses have adequately addressed your concerns, and we kindly request you to reconsider the evaluation in light of these points.

---

> > ### Comment · Reviewer_9Zk2 · 2023-08-12
> >
> > Thank you for the update and clarification. My previous concerns are almost resolved!
> >
> > While the authors have addressed the updates to the questions, I am still pondering the novelty of the cover in this work. To me, it appears to be an application of EGNN combined with Graph Neural ODE using the Newton–Cotes integrator.
> >
> > I want to emphasize that I am not attempting to diminish the value of the authors' work; I genuinely appreciate the specific utilization of algorithmic components such as EGNN and GDEs with Newton-Cotes integration. However, the current manuscript does not sufficiently explain why these algorithmic choices are necessary or more advantageous compared to the existing components. Could you share your opinion on this?

---

> > > ### Author Response · Authors · 2023-08-12
> > >
> > > Thank you very much for your prompt and kind response. We appreciate the opportunity to provide further clarifications.
> > >
> > > We are pleased to provide a more detailed explanation. As mentioned in Lines 50-55,  NC uses a fixed step size and evenly divides the interval into subintervals. Using NC is necessary because our data is organized in the form of trajectories with fixed intervals between states. In such cases, applying Runge-Kutta (rk4 and rk45 as mentioned by the reviewer) and other algorithms becomes challenging. Take the 4th order Runge-Kutta as an example, our problem can be modeled as:
> > >
> > > $x^T = x^0 + \frac{1}{6} (k_1 + 2k_2 + 2k_3 + k_4)$
> > >
> > > where
> > >
> > > $k_1 =  v(t^0, x^0)$,
> > >
> > > $k_2 = v(t^0+\frac{1}{2} T, x^0 + T \frac{k_1}{2})$,
> > >
> > > …
> > >
> > > It is worth noting that the true estimation of $v(t^0+\frac{1}{2} T, x^0 + T \frac{k_1}{2})$ is not feasible since $x^0 + T \frac{k_1}{2}$ is usually not an integer. Additionally, the derivative $v$ is unknown and can be only observed at certain points with fixed intervals. Therefore, we cannot directly apply rk4 to our task.
> > >
> > > From our perspective, GDEs aim to learn the parameterized derivatives and provide estimations at any time point, which requires much more training data to capture the dynamics over time. By contrast, the neural-based methods for reasoning dynamics are designed to “skip” the fine-grained steps in order to reduce the computational costs within small intervals.

---

### Official Review · Reviewer_Gu3W · 2023-07-05

**Soundness:** 4 excellent
**Presentation:** 3 good
**Contribution:** 4 excellent
**Rating:** 8
**Confidence:** 3

**Summary:**

This paper builds on recent advances in deep learning for physical simulation to produce reliable results of long-term consecutive predictions. The authors provide a proof that equivariant graph neural networks learn a linear map from initial velocity to a target velocity, and propose to inject inductive bias based on Newton-Cotes (NC) formulas as alternative of standard equivariant graph convolution. The paper introduces techniques for training, such as a new objective and a new regularization loss for higher order features, and also gives theoretical guarantee that variance can be reduced by increasing the degree of NC formula. Experiments in various multiple body interaction systems demonstrate that the proposed model outperforms  state-of-the-art baselines .

**Strengths:**

The paper is appropriately placed in the current growing literature on scientific machine learning to create a novel fast model for tackling complex problems in the realm of particle systems. Although the proposed architecture is simple in appearance, the motivation of introducing NC formulas is convincing giving both concise visualization and theoretical claims on the limitation of the existing models. Results of the experiments show that introducing NC formulas is effective to perform long-term accurate prediction.

In my opinion this is a good paper and above the acceptance threshold.

**Weaknesses:**

Although the proposed methods NC(k) and NC(k)+ outperforms state-of-the-art baselines, it is still somewhat unclear which components of NC(k)+ help to improve the performance of NC(k)+. What if setting all the coefficients of Newton-Cotes formula to be 1 (or some constant) while remaining the other parameters same? Without some additional ablation study, it is a bit difficult to get a clear picture of how new components of the proposed method would help improving its prediction performance.

In the forward process, how is the coefficients of Newton-Cotes formula obtained? Is the coefficients trainable or derived from the equation (15)? It is also unclear how to derive coordinate at intermediate time step $t^{k}$ once velocity is updated by the formula (13), while the formula (12) is used only when computing the final state at T.

For training time shown in Figure 2, why does the training cost of NC with GMN grow much faster than NC with the other backbones? It will be helpful to get more insight of the trends. Also, the order of NC(k) seems to be set K<=8, due to the catastrophic Runge’s phenomenon. Are there any limitation on the length of time interval [0, T] to which new NC points are spaced? How does the length of the interval affects the performance?

**Questions:**

Please have a look at the weakness mentioned in the strength and weakness section and address these. Overall the idea is interesting, the authors need to substantiate their claims in light of possibly a few more experiments.

**Minor question**

What does “steps” of Table 4 in Appendix C mean?

**Limitations:**

Yes, the limitations are discussed in the appendix.

---

> ### Author Rebuttal · Authors · 2023-08-08
>
> Many thanks for your helpful feedback and suggestions. We will carefully incorporate them into our paper.
>
> **Weaknesses:**
>
> **1. Performing a comparison with a variant NC model with constant coefficients may be helpful to better understand the proposed method.**
>
>   Great idea. We implement a new variant where the coefficients are set to 1 and other hyper-parameters remain unchanged. We report the results as follows:
>
> | Method  |  Intermediate Velocity Error | Final Prediction Error  |
> |---|---:|---:|
> | GMN  | -  | 10.14  |
> | NC (coefficients=1)  | 100.50  | 9.64  |
> | NC   | 6.37  | 9.50  |
> |  NC+ (coefficients=1)  |  10.53 | 9.91  |
> |  NC+   | 0.73  | 9.40  |
>
>   For NC, the model does not implicitly fit the NC formulas when we set all coefficients to 1.  As we can observe, it obtains a large intermediate velocity error. In this case, the model may find a new intermediate value to fit the new coefficients.  It still performs better than the base model GMN, but worse than the original NC. According to NC formulas, if the initial and final points are set, the optimal intermediate points and their coefficients are also determined. Setting all coefficients to 1 may be a worse hint to the model.
>
>   For NC+, we set all coefficients to 1 and compel it to learn the original intermediate velocity. We can see that there is a contradiction between these two objectives. The model performed worse on either prediction.
> We also report the results on the other two datasets in Table 2 in the rebuttal PDF. The proposed method with constant coefficients performs slightly better than GMN on the N-body (1, 2, 0) dataset, but fails to reason dynamics on the more complicated human motion dataset.
>
>   We also report the results on the other two datasets in Table 2 in the rebuttal PDF. The proposed method with constant coefficients performs slightly better than GMN on the N-body (1, 2, 0) dataset, but fails to reason dynamics on the more complicated human motion dataset.
>
> **2. In the forward process, how to obtain the coefficients of Newton-Cotes formula,  are the coefficients trainable or derived from Equation (15)? How to update the coordinate at intermediate time step once velocity is updated by Equation (13).**
>
>   The coefficients are derived from Equation (15) and are hence not trainable. We calculate the intermediate coordinate by adding the predicted velocity to the last coordinate during the inference phase. In Equations (12-13), we can directly feed NC+ with true preceding coordinates during the training phase (a common trick in training sequence models).
>
> **3. Why does the training cost of NC with GMN grow much faster than NC with the other backbones in Figure 2? Also, the order of NC(k) seems to be set K<=8, due to the catastrophic Runge’s phenomenon. Are there any limitations on the length of time interval [0, T] to which new NC points are spaced? How does the length of the interval affect the performance?**
>
>   Regarding the training cost, the GMN model generally requires more time compared to others (e.g., 0.8s per update v.s. 0.6 per update on N-body (1, 2, 0)). It considers the substructure (e.g., hinges and chains) within each object, which yields more cost when we consider more points. In fact, the training cost of GMN doesn’t grow so much faster than that of others. Please note that the y-ticks don’t always start from 0 in Figure 2 (for better readability).
>
>   Regarding the order of NC (k), the neural model learns to compensate for the prediction error (as illustrated in Fig. 1 and Lines 36-44). Setting a larger k may reduce the variance of the prediction error and make the optimization easier, but this strategy does not always work especially when the model is robust. Furthermore, the accuracy gain exponentially decreases as we estimate more points in numerical integration. Take Fig. 1 as an example, a shift from 1e-1 to 1e-2 yields a more significant gain compared to a shift from 1e-3 to 1e-4.
>
> We will clarify these aspects further in the revision.
>
> **Questions:**
>
> **1. Overall the idea is interesting, the authors need to substantiate their claims in light of possibly a few more experiments.**
>
> We appreciate your constructive feedback and hope that our responses have satisfactorily addressed your concerns.
>
> **2. Minor question -What does “steps” of Table 4 in Appendix C mean?**
>
> Sorry for the confusion. “# steps” is a shorthand for “the number of estimation steps”, i.e., k in our main paper. We have corrected it for better clarity.

---

> > ### Comment · Reviewer_Gu3W · 2023-08-11
> > **Additional questions**
> >
> > Thank the authors for the response. Most of my concerns were addressed. I have a few follow-up questions on the following response:
> >
> > >Regarding the order of NC (k), the neural model learns to compensate for the prediction error (as illustrated in Fig. 1 and Lines 36-44). Setting a larger k may reduce the variance of the prediction error and make the optimization easier, but this strategy does not always work especially when the model is robust. Furthermore, the accuracy gain exponentially decreases as we estimate more points in numerical integration. Take Fig. 1 as an example, a shift from 1e-1 to 1e-2 yields a more significant gain compared to a shift from 1e-3 to 1e-4.
> >
> > For
> >
> > >Setting a larger k may reduce the variance of the prediction error and make the optimization easier...
> >
> > my question is the impact on prediction error when making the interval length longer, say [0, T] to [0, N*T]. Since the order of NC(k) (and NC(k)+) is bounded from above due to the catastrophic Runge’s phenomenon, the prediction error would also be bounded from the bottom. It would be helpful if the authors provide insights or theoretical/experimental results that clarify the robustness of the proposed method against the interval length.
> >
> > > Take Fig. 1 as an example, a shift from 1e-1 to 1e-2 yields a more significant gain compared to a shift from 1e-3 to 1e-4.
> >
> > What do you mean by shift? In appearance, I don’t see any terms relating to shift in figure 1. Can you expand the explanation?

---

> > > ### Author Response · Authors · 2023-08-11
> > >
> > > Thank you very much for your prompt response. We appreciate the opportunity to provide further clarifications.
> > >
> > > **1. The true question is the impact on prediction error when making the interval length longer, say [0, T] to [0, N*T].**
> > >
> > > We apologize for the oversight. Exploring the impact of the interval length $T$ is indeed an interesting problem. It is important to note that a larger $T$ implies predicting the future states more distant from the input, posing greater difficulty for the model. Our theoretical analysis (Equation (11), Proposition 3.2) also reveals a positive correlation between the variance/error of prediction and the interval $T$. We have conducted some new experiments and present the results (NC ($k=2$) without additional training data) as follow:
> > >
> > >
> > > |  $T$ | N-body (1,2,0) NC/GMN  | MD17 (Aspirin) NC/GMN  | Motion NC/GMN |
> > > |---|---:|---:|---:|
> > > | $0.5T$ | 1.50/1.81 | 6.97/7.32 | 10.2/13.7 |
> > > | $T$ | 1.55/1.84 | 9.50/10.14 | 30.0/43.9 |
> > > | $2T$ | 1.59/1.98 | 14.93/16.60 | 54.4/86.1 |
> > > | $4T$ | 1.62/2.10 | 19.05/21.60 | 98.6/165.8 |
> > >
> > > We can observe that the choice of $T$ determines the difficulty of the task to some extent. The performance of both methods decreases as $T$ increases. Specifically, the performance of the two methods slightly decreases on the N-body dataset due to its relative simplicity, but drops more rapidly on the MD17 and Motion datasets. Nevertheless, our method consistently and significantly outperforms the baseline across different $T$ values and datasets, providing empirical evidence of its robustness.
> > >
> > > Additionally, we have conducted an experiment to evaluate the performance of our method w.r.t $k$ under a long interval ($4T$). The results on MD17 (Aspirin) are presented as follows:
> > >
> > > |  Method | $T$  | $4T$ |
> > > |---|:---:|:---:|
> > > | GMN  | 10.14  |  21.60 |
> > > | NC ($k=2$)  | 9.50  | 19.05  |
> > > | NC ($k=4$)  | 9.34  | 18.69  |
> > > | NC ($k=8$)  | 9.49  | 18.61  |
> > >
> > > We can observe that increasing $k$ provides more improvement with longer $T$, but the gain (4→8) remains relatively marginal.
> > >
> > > **2. In appearance, I don’t see any terms relating to shift in Figure 1. Can you expand the explanation?**
> > >
> > > We are pleased to provide a more detailed explanation. The higher-order NC formulas offer improved accuracy as well as smaller error terms. Specifically, the error terms are proportional to $T^3$, $T^5$, $T^7$ for $k=1$, $2$, and $4$, respectively. It is worth noting that the numerical gain in accuracy diminishes exponentially as $k$ increases. This can be roughly understood as $T^3 - T^5 \gg T^5 - T^7$. In other words, the incremental improvement in accuracy becomes progressively smaller with higher-order NC.
> > >
> > > Take Figure 1 as an example, the blue + yellow area represents the error of NC ($k=0$), which is significantly larger than the error of NC ($k=1$) (the blue area). The yellow area precisely represents the accuracy gain. If we extend the estimation to NC ($k=2$) in the figure, the area of its error must be a sub-area of the error of NC ($k=1$). Consequently, the accuracy gain from $k=1$ to $k=2$ is also a sub-area of the blue area and significantly smaller than the yellow area.

---

> > > > ### Comment · Reviewer_Gu3W · 2023-08-12
> > > >
> > > > Thank you very much for addressing the additional comments.
> > > >
> > > > > It is important to note that a larger $T$ implies predicting the future states more distant from the input, posing greater difficulty for the model.
> > > >
> > > > Yes, I agree with the opinion, this is one of the fundamental issues of surrogate models. But I still believe that providing results of the additional two experiments as well as theoretical results such as Proposition 3.2 and 3.3 is quite beneficial for readers working on this domain to get a clear picture of the limitation of the proposed method.
> > > >
> > > > In light of the further clarification on the limitation as well as the novelty of the proposed model, I increased my score. The authors’ rebuttal is satisfactory to my concerns and strengthened their claims further.
> > > >
> > > > Finally let me add comments on Proposition 3.2:
> > > >
> > > > * Expressing the equation (5) using the notation $\epsilon^{2}_{NC(0)}$ without any explanation is confusing for readers, because the notation seems to have not been introduced prior to this equation, and even looks like a constant.
> > > >
> > > > * Should $\epsilon_{NC}(0)$ in the equation (10) be $\epsilon_{NC(0)}$?
> > > >
> > > > I appreciate for the authors’ all efforts during the rebuttal period. I really enjoyed the discussion!

---

> > > > > ### Author Response · Authors · 2023-08-12
> > > > >
> > > > > We sincerely appreciate your valuable feedback and unwavering support throughout the review process. We have provided an explanation for $\epsilon_{\text{NC}(0)}^2$ and corrected the typo in $\epsilon_{\text{NC}(0)}$ in the revision. We are truly grateful for your contribution.

---

### Official Review · Reviewer_MNf4 · 2023-07-06

**Soundness:** 3 good
**Presentation:** 3 good
**Contribution:** 3 good
**Rating:** 8
**Confidence:** 4

**Summary:**

This paper integrates Newton-Cotes formulas from numerical analysis in (equivariant) graph networks. The authors introduce an approach that makes several velocity estimations to predict the integration towards the final state of a system.
Further, they show that standard equivariant graph networks can be interpreted as a trivial case of their formalism. The proposed method (NC) has provably lower errors/variance, and becomes better with more velocity estimations. NC can be used with supervision from intermediate velocities, but it can also work well without supervision.
The authors demonstrate the effectiveness of their method on several benchmarks, and using multiple graph network architectures as a backbone.

**Strengths:**

The paper is very well-written and easy to follow. The proposed method is well-motivated from numerical analysis, and it is simple and clear.
The proposed method shows improvements on 3 different benchmarks, and using 3 established equivariant graph networks as a backbone.
The non-supervised version of the method (NC) can be directly integrated in many existing graph network architectures.

**Weaknesses:**

The analysis on the prediction errors of intermediate velocities (figures 3, 6, 7) could be more in-depth, especially for the unsupervised version of the method (NC). It is unclear why the errors often increase as training progresses, or why they tend to fluctuate erratically, and whether this has an impact on performance, or if the model can make up for it.

The qualitative visualizations, especially for MD17 and n-body (figure 8) are very hard to interpret.

**Questions:**

1. Following the weaknesses listed above, in figures 3 and 6, why do the errors fluctuate or keep increasing with epochs? What is its impact on performance?

2. The Newton-Cotes formulas exhibit a reflection symmetry. Is it possible that the model is learning to predict the reflected intermediate velocities? An ablation study on that would be useful.

3. In subsection 4.6, the authors perform an ablation study on the method's ability to make consecutive predictions. There are related works that propose equivariant graph networks for multiple step prediction, including LoCS [2] and EqMotion [1]. It would be interesting to examine the applicability and added value of NC to such methods.
As an example, it would be interesting to see the integration of NC in LoCS, since this work treats velocities in a principled manner and makes predictions in local frames.


#### Minor

4. In lines 161-162, the authors mention that "EGNN always uses the same input velocity and coordinate as constant features in different layers". However, EGNN does use the updated coordinates in different layers, at least in the n-body setting.

#### References
[1] Xu, Chenxin et al. EqMotion: Equivariant Multi-agent Motion Prediction with Invariant Interaction Reasoning. CVPR 2023.

[2] Kofinas, Miltiadis et al. Roto-translated Local Coordinate Frames for Interacting Dynamical Systems. NeurIPS 2021.


**Limitations:**

The authors have adequately addressed the limitations of their work.

---

> ### Author Rebuttal · Authors · 2023-08-08
>
> We are grateful for your encouraging comments and insightful observations. We hope the following responses address your concerns:
>
> **Weaknesses:**
>
> **1. The analysis on the prediction errors of intermediate velocities (figures 3, 6, 7) could be more in-depth, why the error of the unsupervised NC tends to fluctuate erratically, and whether this has an impact on performance, or if the model can make up for it.**
>
>   Thanks very much for your suggestion. To ensure visual clarity, we have magnified the prediction errors (%), which may make the unsupervised errors seem more erratic than they truly are; otherwise, the error of the supervised version would look like a zero line.
>
>   We provide a new figure (Figure 1 in the rebuttal PDF) depicting both the intermediate error and final prediction error. Despite the fluctuations, the intermediate error’s impact on the final prediction error is minimal, as the neural model compensates accordingly.
>
>   The unsupervised NC does not have an explicit loss to minimize the intermediate error (as stated in Lines 261-264). We want to use this experiment to illustrate that even without an explicit supervised loss, the model automatically adjusts to fit the intermediate points to obtain a better prediction for the final states.
>
> **2. The qualitative visualizations for MD17 and n-body (figure 8) could be more readable.**
>
>   Thanks for your suggestion. The input objects are molecules in MD17, but the base model GMN considers only a subset of chemical bonds during training and inference. This is why the objects in the figures do not look like molecules. We have updated the illustrations and included a new figure (Figure 2 in the rebuttal PDF). For the N-body figures, presenting a more readable version is challenging as the input object is a mixture of points, sticks, and hinges.
>
> **Questions:**
>
> **1. Following Weakness 1, why do the errors fluctuate or keep increasing with epoch and what is its impact on performance?**
>
> Please see our response to **Weakness 1**.
>
> **2. Is it possible that the model is learning to predict the reflected intermediate velocities?**
>
>   Thanks for your suggestion. It is indeed possible to predict the intermediate velocity given the initial and final states. We perform a comparison with a GMN variant where we concatenate the outputs of the initial and final states to predict the intermediate velocity. The results on the motion dataset are shown in the table below:
>
> |  Method | Intermediate Velocity Prediction  | Final State Prediction  |
> |---|---:|---:|
> | GMN  | 5.4  |  43.9 |
> | NC  | 2.8  | 29.6  |
>
> We can observe that our method employing NC formulas to inversely infer intermediate velocities significantly outperforms the GMN variant. A more interesting and challenging task may be predicting the continual trajectories between the initial and final states, which we intend to explore in the future.
>
> **3. There are EGNN methods for multiple step prediction, including LoCS [2] and EqMotion [1]. It would be interesting to examine the applicability and added value of NC to such methods.**
>
> We have applied our method to EqMotion and obtained some preliminary results. To ensure a fair comparison, we use identical hyper-parameter settings and we do not employ any additional data. The results are shown in the following table (ADE and FDE denote average displacement error and final displacement error, respectively):
>
> | Method  |  N-body Prediction ADE | N-body Reasoning Accuracy  | MD17 (Aspirin) ADE/FDE  | MD17 (Benzene) ADE/FDE  | MD17 (Ethanol) ADE/FDE  | MD17 (Malonaldehyde) ADE/FDE |
> |---|---|---|---|---|---|---|
> | EqMotion | 16.8  | 97.8  | 5.95/8.38  | 1.18/1.73 | 5.05/7.02 | 5.85/9.02 |
> | NC (EqMotion)  | **16.2**  | **98.9**  | **4.74/6.76**  | **1.15/1.63** | **5.02/6.87** | **5.19/8.07** |
>
>
> Clearly, our NC (EqMotion) outperforms the base model EqMotion on all datasets and across all metrics. We will also release the source code of NC (EqMotion) if the paper gets accepted. Thank you again for your insightful comment.
>
>
> **4. Minor - In lines 161-162, the introduction to EGNN should be revised. It uses the updated coordinates in different layers at least in the n-body setting.**
>
> Many thanks. We have corrected it.

---

> > ### Comment · Reviewer_MNf4 · 2023-08-17
> > **Official Comment by Reviewer MNf4**
> >
> > I would like to thank the authors for their rebuttal; they have addressed my questions. I am very positive towards this paper and I think its contribution is significant. I am keeping my score to an 8. I hope the authors include the related works on multi-step prediction in the camera-ready version, alongside any experiments performed with the proposed method on top of them.

---

> > > ### Author Response · Authors · 2023-08-17
> > >
> > > We are truly grateful for your valuable comments and unwavering support. We will update the relation work and experiment sections in our revision to include the multi-step prediction methods and corresponding results. Thank you once again for your insightful suggestions.

---

### Official Review · Reviewer_P4zh · 2023-07-24

**Soundness:** 4 excellent
**Presentation:** 3 good
**Contribution:** 3 good
**Rating:** 6
**Confidence:** 3

**Summary:**

The paper introduces a new method for predicting system dynamics. Unlike past works that predict the average velocity between start and end time, the proposed approach makes several velocity estimation at different time points and then uses Newton-Cotes formula for computing the integration. This method reduces the prediction error and improve accuracy in predicting future state of the system. The paper improves this method further by incorporating loss for each predicted velocity estimate, and shows better performance than current SOTA systems.

**Strengths:**

1. The paper is clearly written with good figures. Supplementary material provides additional GIFs and proofs to help with understanding.
2. The paper proposes an interesting method to improve the current dynamics prediction systems by predicting velocity estimates at different time points and then using Newton-Cotes method to integrate them. Paper proofs that this method leads to reduction in variance, and therefore the error. NC+ approach further reduces the error by incorporating additional loss.
3. The paper shows that even their NC method outperforms the previous SOTA approaches significantly.
4. Despite predicting velocities at different time points, this method doesn't degrade the training time a lotl.

**Weaknesses:**

I feel the novelty of the method is limited. The method replaces the average velocity estimate with multiple velocity estimates that are then integrated over, something that seems pretty obvious. Also, something that is not clear is why the performance improvement saturates at only K=2, as shown in Fig 2? Another weakness is that the inference time scales linearly as the value of K is increased.


**Questions:**

I would like authors to address the points raised in weakness section. What is the intuition behind accuracy not increasing past K=2. It seems that higher K should result in even better performance, but that's not the case.

**Limitations:**

The authors have not addressed limitations in their work.

---

> ### Author Rebuttal · Authors · 2023-08-08
>
> Thank you very much for your detailed and insightful comments. We have addressed your concerns below and hope our responses provide clarity:
>
> **Weaknesses:**
>
> **1. The novelty of the method may be limited. The method seems intuitive.**
>
>   The novelty of our paper lies not just in the design of neural models but also in the theoretical exploration of EGNNs. We reformulate the existing EGNN methods in the form of numerical integration, which offers a more comprehensive view of reasoning system dynamics.
>
>   NC (k) is straightforward, yet powerful and fully supported by the theoretical proof. It significantly improves the performance of EGNNs on all datasets. Thanks to its simplicity, the computational complexity is also acceptable. The training time increases slightly as it estimates more points (as shown in Fig. 2).
>
> **2. Why does the performance improvement saturate at K=2, as shown in Fig 2?**
>
>   Sorry for the confusion and we will improve the presentation. As it may not be very clear from Fig. 2, we provide the detailed results in Table 1 in the rebuttal PDF. The prediction error is still reduced (slightly) as we set a larger k. This behavior could be attributed to the following reasons: (1) The neural model learns to compensate for the prediction error (as illustrated in Fig. 1 and Lines 36-44). Setting a larger k may reduce the variance of the prediction error and make the optimization easier, but this strategy does not always work, especially when the model is robust. (2) The accuracy gain exponentially decreases as we estimate more points in numerical integration. Take Fig. 1 as an example, a shift from 1e-1 to 1e-2 yields a more significant gain compared to a shift from 1e-3 to 1e-4. (3) Runge’s oscillation phenomenon (Lines 152-154) tells us that more interpolation points may not always lead to more accurate results.
>
> **3. The inference time scales linearly with the increase in k. The authors haven't addressed this limitation.**
>
>   There seems to be a misunderstanding. As shown in Fig. 2 (rows 2 and 3), the training time only increases slightly. We provide the detailed training time on N-body (1, 2, 0) below:
>
> | k  | GMN   | EGNN   | RF   |
> |---|:---:|:---:|:---:|
> | 0  | 1817.0   | 1355.4   |  1328.2  |
> | 1  |  1800.9 | 1357.4  |  1333.2 |
> | 2  |  2057.9 | 1395.6  |  1328.9 |
> | 5  |  3749.0 | 1841.2  | 1946.9  |
>
>   Please note that k is in {0, 1, 2, 5}.
>
> **Questions:**
>
> **1. I would like authors to address the points raised in the weakness section.**
>
>   Thank you again for your feedback. We hope our responses to **Weaknesses** have addressed your concern.
>
>
>
> **Limitations:**
>
> **1. The authors have not addressed limitations of their work.**
>
>   We acknowledge this concern and have indeed discussed this issue in Sec. 3.5. As elaborated in our response to Weakness 3, the increased computational cost with higher k values is affordable. We believe this trade-off is worth the advantages offered by our method.

---

> > ### Comment · Reviewer_P4zh · 2023-08-16
> >
> > Thanks for your response and addressing my concerns. I have increased my final score to 6.

---

> > > ### Author Response · Authors · 2023-08-17
> > >
> > > We sincerely appreciate your increased rating and recognition of the efforts we put into addressing your concerns. Your contribution to our work is highly valued and greatly appreciated.

---

### Author Rebuttal · Authors · 2023-08-08

Dear all reviewers:

We sincerely appreciate the time and effort you have dedicated to reviewing our paper. Your comments are invaluable in helping us enhance the quality of our paper.

We would like to thank Reviewers P4zh and MNF4 for their suggestions regarding the readability of some figures. We have addressed their concerns by updating the figure and including some examples in the rebuttal PDF.

We are also grateful to Reviewers Gu3W and 9Zk2 for their suggestions regarding new baselines. We have presented the new experimental results in the rebuttal PDF and our replies.

We especially thank Reviewer MNf4 for the suggestion about applying our method to the recent paper EqMotion [1]. We have conducted experiments and we are pleased to report that some preliminary results are highly promising:

| Method  |  N-body Prediction ADE | N-body Reasoning Accuracy  | MD17 (Aspirin) ADE/FDE  | MD17 (Benzene) ADE/FDE  | MD17 (Ethanol) ADE/FDE  | MD17 (Malonaldehyde) ADE/FDE |
|---|---|---|---|---|---|---|
| EqMotion | 16.8  | 97.8  | 5.95/8.38  | 1.18/1.73 | 5.05/7.02 | 5.85/9.02 |
| NC (EqMotion)  | **16.2**  | **98.9**  | **4.74/6.76**  | **1.15/1.63** | **5.02/6.87** | **5.19/8.07** |

The empirical evidence from these new tasks further validates the effectiveness of our method. We are also committed to releasing the source code of NC (EqMotion) if our paper is accepted.


Best Regards,

Authors

[1] Xu, Chenxin et al. EqMotion: Equivariant Multi-agent Motion Prediction with Invariant Interaction Reasoning. CVPR 2023.

---

### Decision · Program_Chairs · 2023-09-21

**Decision:**

Accept (spotlight)

**Comment:**

The paper proposes a new method called Newton-Cotes Graph Neural Networks (NC) to improve prediction of system dynamics over long time horizons. It uses multiple velocity estimations based on Newton-Cotes numerical integration formulas instead of a single velocity estimate used in prior work.
Several reviewers recommended acceptance, citing the clear writing, novel integration of numerical methods with graph neural networks, strong experimental results, and theoretical analysis. Their main criticisms were around novelty, limitations of the method, and some minor presentation issues.

Reviewer 9Zk2 initially recommended rejection, arguing the method is a special case of a broader class of models (Graph Neural ODEs) that was not compared to.
In their rebuttal, the authors added comparisons to depth-varying Graph Neural ODEs, showing their proposed NC method achieves better performance. They argued their method is complementary to, not a subset of, Neural ODE methods. Reviewer 9Zk2 was partially convinced but still questioned the necessity of using Newton-Cotes formulas specifically.

Through the author rebuttal, more justification for using Newton-Cotes was provided:
- Their data has fixed intervals between states, which makes it straightforward to select intermediate points satisfying Newton-Cotes conditions. This would be more difficult for methods like Runge-Kutta that use adaptive step sizes.
- Newton-Cotes allows them to easily incorporate intermediate supervision through NC+, which helps improve performance.
- The simplicity of Newton-Cotes makes it easy to integrate into existing graph network architectures.

Overall, I believe the paper does make solid contributions experimentally and theoretically. So I recommend acceptance, while encouraging the authors to expand on the rationale for Newton-Cotes in the camera-ready version.